# Comparing Power Supply Technologies for Public Transport Buses through the AHP and the Fuzzy DEMATEL Method

Caterina Caramuta [1,*], Giovanni Longo [1], Elio Padoano [2] and Maria Vesela [3]

1   Department of Economics, Business, Mathematics and Statistics, University of Trieste, 34127 Trieste, Italy; giovanni.longo@dia.units.it
2   Department of Engineering and Architecture, University of Trieste, 34127 Trieste, Italy; padoano@units.it
3   Department of Management and Transport, Dnipro University of Technology, 49000 Dnipro, Ukraine; mves357@gmail.com
*   Correspondence: ccaramuta@units.it

**Abstract:** The selection of power supply technology for buses is a critical task given the increasing attention paid to environmental sustainability in the public transport sector. Indeed, the compliance of vehicle operational requirements with service characteristics is essential to provide users with an efficient offer. To this end, this study investigates the factors affecting such choices by performing two evaluation procedures, with the integration of different techniques and the engagement of an expert panel. The Analytic Hierarchy Process (AHP) method was used to identify the best power supply technology among a few solutions in both procedures, which differed in the number of analyzed criteria. A literature review suggested a wide set of criteria considered in the first assessment, which were then limited to the most influential criteria using the fuzzy DEcision-MAking Trial and Evaluation Laboratory (DEMATEL) method. Notably, the latter enabled the reduction in the number of the criteria owing to the revealing of cause–effect relationships among them. The methodology was applied to a case study in the city of Trieste, Italy, comparing rankings obtained from the two appraisal procedures, which showed the predominance of internal combustion engine buses over hybrid and electric buses in terms of operational and financial aspects, despite their environmental impact.

**Keywords:** public transport; bus power supply technology; AHP; fuzzy DEMATEL

## 1. Introduction

In the modern world, the development of road transport is moving in the direction of using vehicles with zero emissions in response to the severely damaged state of the environment caused by pollution. Indeed, at the European level, the transport sector is estimated to cause almost 25% of the greenhouse gas emissions, and more than 90% of roads vehicles are run by fossil fuels. Consequently, the concentration levels of pollutants in urban areas exceed the threshold values acceptable for human health in nearly all cases [1]. Therefore, users have been led more and more often to employ vehicles using alternative technologies of power supply, like, for instance, electricity [2]. In this regard, the global increase in the number of electric vehicles [3] reflects this trend.

Road collective transport is an integral part of the development of modern cities, enabling people to satisfy their mobility needs in the presence of high demand volumes. However, considering the harmful effects of transport on the environment and society, transport operators are requested to offer services that implement the latest technologies, with the aim of improving service quality and efficiency while using sustainable vehicles. In line with this, the process of providing electromobility services has already been actively implemented in many European countries and it is intended to be accelerated to achieve sustainability goals in a short time [4].

The selection of the power supply technology for buses is determined by many factors, ranging from environmental, social, and financial aspects [3] to operational, technological, and strategic aspects [5], as illustrated in Figure 1.

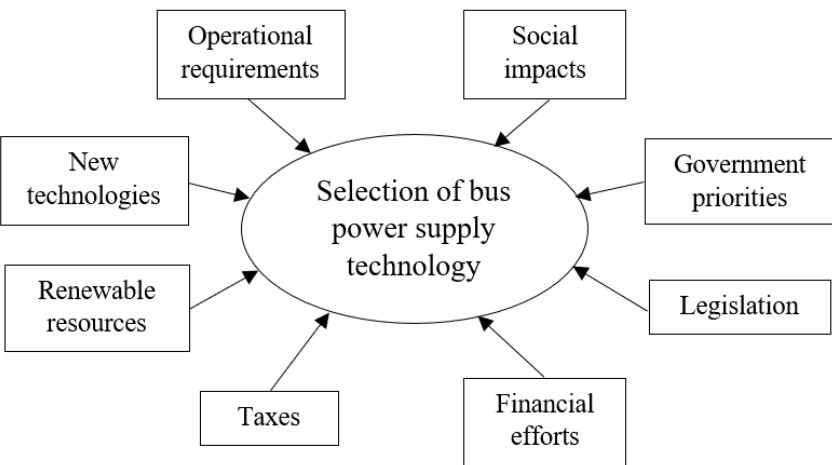

**Figure 1.** Aspects influencing the selection of the bus power supply technology.

Such a task represents one of the main stages of the replacement of internal combustion engine vehicles (ICEVs) with alternatively fueled ones and entails decisions with respect to different issues concerning, for example, the type of battery and the positioning of charging stations. All these features contribute not only to the performance of the resulting transport service but also to the definition of capital and running costs and, ultimately, of mobility plans. As a matter of fact, considering that a few years ago electric buses turned out to have various drawbacks when compared to ICEVs, they now have become quite competitive in terms of possible mileage, expenses, and risks, depending on local tax requirements and operating costs.

With respect to the available techniques supporting decision-making processes, the socio-technical nature of transport problems [6] implies resorting to specific evaluation methods that include both quantitative and qualitative aspects. In addition, the possible presence of interdependent relationships among such aspects definitely increases the complexity of evaluation procedures, which must account for the mutual influence of aspects other than their level of importance. A clear distinction between these two concepts is essential for analyzing the potential of existing decision-making methods and to guide the selection of the most appropriate approach. The adoption of advanced methodologies incorporating more than one assessment technique constitutes a common solution to embrace the dependencies and priorities of evaluation aspects, owing to the compensation of the limitations of the involved techniques.

In line with the goal of assisting public transport operators in the transition towards the provision of a more environment-friendly offer, this paper proposes a comparison between the rankings of alternative bus power supply technologies obtained from two different assessment approaches. Indeed, the same set of alternatives was evaluated through the Analytic Hierarchy Process (AHP) method, but a diverse set of criteria was considered in the two cases. In the first assessment, a wide set of criteria, covering different categories of factors, was employed. This set was reduced in the second assessment thanks to the application of the fuzzy DEcision-MAking Trial and Evaluation Laboratory (DEMATEL) method. The combination of these two techniques enabled the evaluation of the most influential criteria, which originated from the identification of cause–effect relationships through the fuzzy DEMATEL method. The reasons for the implementation of the considered evaluation techniques are, first, the acknowledged suitability of the AHP method for multi-faceted transport problems, which is demonstrated by the fact that it has been the most popular multi-criteria decision-making method used in the transport sector

worldwide in the first two decades of the 2000s [7]. Second, the motivation for using the DEMATEL method has been its diffused employment to analyze interdependencies among factors for different transport-related issues, including electric mobility and supply chain management, as reported in [8–13]. Ultimately, the developed methodology has served a two-fold objective: on the one hand, facilitating the appraisal procedure by reducing the set of evaluation criteria and, on the other hand, providing insights into the difference between criteria influence and criteria significance on the decision recommendation. The criteria were selected mainly through a literature review of the principal factors affecting the selection of the bus power supply technology based on a variety of aspects. In addition, the entire assessment process was performed by engaging multiple actors, each with different expertise and nationality, who participated in bringing their own perspective. The proposed methodology was applied to a case study of the Italian city of Trieste to support the local public transport company in addressing the problem of fleet replacement.

Driven by the significant environmental and social impact of the transport sector, the study reported in this paper represents an attempt to fill the research gap concerning the adoption of an integrated methodology to evaluate both the mutual influence and the relative importance of aspects related to the selection of the best power supply technology for public transport buses. To the best of the authors' knowledge, the current scientific literature lacks similar contributions, although the need and subsequent tendency for a greater use of alternative-fueled vehicles have been demonstrated by the growing number of investigations on the topic in recent years. Furthermore, insights from the real-world implementation of the developed evaluation methodology constitute an added value of this study, which translates into practical decision-making support for transport operators. Indeed, the analysis performed on the ranking of alternatives is meant to shed light on the convenience and practicability of replacing ICEVs with more environmentally sustainable solutions.

The remainder of this paper is organized as follows. Section 2 includes a literature review of the principal factors affecting the selection of the technological solution for bus power supply technology, which includes social, transport, environmental, operational, and economic aspects. The outcomes of such an analysis constitute the starting point of the suggested evaluation methodology, which is explained in Section 3 by describing the two employed techniques and their integration. Section 4 illustrates the application of the methodology to a case study, the results of which are described and discussed in Section 5. Finally, the conclusions are reported in Section 6, which synthesizes the main advantages of the methodology and its possible advancements.

## 2. Literature Review

As mentioned, decisions concerning the selection of the technological solution for bus power supply technology consider multiple aspects, including social, transport, environmental, operational, and economic factors. This is motivated by the fact that the public transport service is taken into account as a whole: a representation of the relationship among its components is reported in Figure 2 referring to the "Driver-Bus-Road-Environment" (DBRE) system.

Previous studies have used such factors not only as parameters to solve the problem of substituting the bus fleet in existing transport systems but also in studies on risk management [14]. In other studies [14,15], a comparison between groups of these criteria was proposed.

In this study, the main factors affecting the selection of bus power supply technology were identified based on a literature review [3,16–26] in the field and on the personal experience of the experts involved in the study in the fields of transport planning, operations, and technology. The considered factors were collected according to the categories described below.

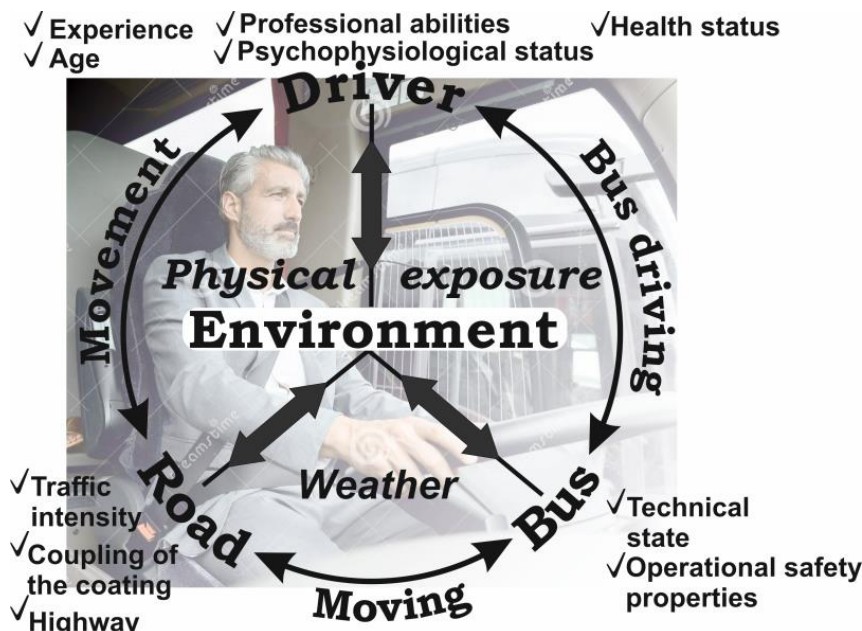

**Figure 2.** "Driver-Bus-Road-Environment" (DBRE) system [Source: own elaboration].

Social factors. The driver is one of the main elements of the system, since the effectiveness of the latter factor depends on his/her professionalism, the conditions of the working environment, the satisfaction level of working conditions, and, most importantly, on-board safety.

Operating factors. The second element of the DBRE system is the vehicle, i.e., the bus. In [14], researchers highlighted the importance of parameters characterizing public transport vehicles and lines, which include, but are not limited to, the bus length, the line frequency, the number of scheduled bus stops, the line route to the city center, and the access to the power grid at the terminus stops. In this regard, the literature review reported in [19] identifies relevant transport factors such as speed, passenger capacity, maximum power, and, in the case of electric vehicles, battery capacity and charging time.

Transport factors. The third element of the DBRE system is the road, i.e., the infrastructure. The efficiency of bus passenger transport is regulated by indicators such as passenger flow, which determines the bus capacity, and the length of the route, which determines the entity and the parameters of the power source. Further aspects to be considered refer to the arrangement of lanes dedicated to bus transit as well as the prioritization of this latter factor over other traffic components in case of congestion, which both contribute to provide an efficient service and to limit power wasting. According to the authors' experience, in addition to these factors, the availability of modern charging stations is significant because the charging time depends on the type of charging station, as discussed in [20].

Environmental factors. As mentioned in [27], environment-friendly buses contribute to reducing air pollution since they are powered by sustainable resources rather than fossil fuels. They have zero tailpipe emissions, including nitrogen oxide and particulate matter. Another significant advantage of these vehicles is that they emit less noise, which is of particular importance in residential areas. The relevance of the environmental impact of transport and its minimization is underlined in [28]. Temperature and climate need to be considered when selecting bus power supply technologies, as they can directly impact operations and, thus, affect the range of possible vehicle typologies to choose from. This relationship is well demonstrated in [21], in which the authors discussed the dependence of the autonomy of electric buses on atmospheric conditions.

Economic/financial factors. The implementation of modern transport technologies for passenger transport and the effectiveness of the proposed measures are often evaluated through economic factors, which are among the most significant components for assessing

the performance of transport companies. Notably, the following two indicators are of great importance at the financial level: the amount of invested funds and their payback period. For the public, the perception of the realized measures is also influenced by their cost, which is reflected in the expenses of users moving in space. A technical and economic comparison of different electric buses is widely described in [20], along with possible scenarios of use.

With regard to the methodological approach, a survey of state-of-the-art evaluation techniques employed to select non-conventional power supply technologies for buses was conducted, focusing on contributions that implement the AHP and DEMATEL methods in a combined manner. In this respect, it seems that no previous investigations on such a topic have considered the integration of AHP and DEMATEL, but some attempts have been made to address the decision problem using, alternatively, one of the two methods in conjunction with other assessment techniques. For instance, in [29] the authors took advantage of the integration of fuzzy AHP and fuzzy VIKOR (VIekriterijumsko KOmpromisno Rangiranje) to determine the weights of the criteria and rank the alternatives for the selection of vehicles for public transport. The applicability of the developed methodology was tested in Ankara, Turkey, by actively engaging experts in the evaluation procedure and performing a sensitivity analysis to discuss possible variations in the decision recommendation. An analogous approach was adopted in [30], with the only difference being that the outcomes obtained using the VIKOR method were compared with those resulting from the application of the Technique for Order Preference by Similarity to Ideal Solution (TOPSIS) method. In this case, the identification of the best compromise alternative fuel mode served in the decision-making process for the urban areas of Taiwan. Similarly, in [31], the Interval Type-2 fuzzy AHP and TOPSIS were used to appraise bus alternatives for public transport in Istanbul, Turkey, to tackle the uncertainties that characterize real-world problems. Unlike previous contributions, the evaluation criteria and alternatives were chosen using the Delphi method, while the priorities of the former and the ranking of the latter were defined through the fuzzy AHP and TOPSIS. From a long-term perspective, the authors of [32] developed a hybrid life cycle sustainability assessment model that enables the quantification of the environmental, economic, and social impacts of alternative-fueled buses, and then assists the ranking of the different technological solutions on the basis of their relative sustainability performance. In order to accomplish this latter task, the Interval-Valued Neutrosophic Fuzzy (IVNF)-AHP has been integrated with the Combined Compromised Solution (CoCoSo) method to investigate the transition towards net-zero transport systems in Qatar. With regard to the DEMATEL method, researchers in [33] applied this technique to compute the weights of the evaluation criteria used to select the best alternative fuel for buses and reach the ultimate goal of limiting greenhouse gas emissions. Notably, the application of the DEMATEL method was functional to the subsequent ranking of the alternatives, which was obtained by means of the integration with the COmplex PRoportional ASsessment (COPRAS) method.

The scientific papers examined in the literature review specifically dedicated to the application of AHP and DEMATEL methods for the selection of alternative-fueled buses confirmed the categories of the factors reported above as the main parameter domains affecting such a decision. Proof of this is the fact that the analyzed contributions encompass aspects related not only to environmental pollution, financial expenses, and vehicle and infrastructural requirements but also to human health and comfort. However, the study included in this paper marks a step forward with respect to the existing ones because, in addition to providing the level of importance of the examined criteria, it investigates their mutual influence. Therefore, this study adds a further dimension of analysis to the evaluation procedure, fostering a focus on the criteria that affect the assessment.

Overall, based on the literature review and the expertise of experts in the field of public transport and electric vehicles, a set of factors were chosen to rank different alternative bus power supply technologies. The details of the theoretical foundation of the developed evaluation methodology and its implementation are explained in the following section,

together with its application to a case study. As such, the present study contributes to overcome the following knowledge gaps, highlighted in [32]:

- The limitedness of the application field of previous investigations to small-sized vehicles, extending the assessment of alternative vehicle technologies to city buses;
- The need to cover uncertainty in decision-making processes, thanks to the use of a fuzzy approach for the management of different expert judgments.

## 3. Methodology

To identify the best power supply technology for buses with respect to various aspects, an evaluation methodology was developed using different techniques, as graphically explained in Figure 3. Following the object definition, the initial stage of the methodology consisted of performing a literature review on the factors affecting the selection of the power supply technological solution for buses. Such factors have served as evaluation criteria for the consequent assessment procedure, which was first performed by adopting the AHP method and then combining it with the fuzzy DEMATEL method. The former application of the AHP method considered a list of all the factors derived from the literature review, which was integrated with additional factors based on the experience of the involved experts. The related ranking of alternatives was computed according to the preferences of the panel of experts, with reference to the priority of the criteria and performance of the alternatives. However, as anticipated, the difficulty in managing a set of numerous evaluation criteria led to the need to resort to a rigorous approach to reducing them, which, in this study, is represented by the combination of the AHP and DEMATEL methods. More specifically, the fuzzy version of the DEMATEL method was used to capture cause–effect relationships among the selected factors in order to distinguish influential factors from influenced factors [16,34,35]. Thus, a second AHP-aided evaluation was performed including only the influential factors suggested by the application of the fuzzy DEMATEL method based on the judgments expressed by the same group of experts. Finally, a comparison between the ranking of alternatives considering the wider set of criteria and the reduced set was performed. Therefore, the integration of the two techniques has been meant not only to reduce the number of analyzed criteria, and, consequently, the fatigue experienced by experts while responding to pairwise comparisons among criteria, but also to investigate the potential variations in the ranking of alternatives depending on the adopted evaluation approach.

The AHP method is a well-known multi-criteria evaluation method which was created by Professor T. L. Saaty in the 1970s [36,37]. It considers the decomposition of the decision problem into simpler parts according to a hierarchical structure in which the elements are pairwise compared in order to define their relative priority. The elements contained in the different levels of the hierarchy represent, from top to bottom, the main goal of the evaluation, the assessment criteria according to which the performances of decision alternatives are assessed, and, finally, the decision alternatives. Each element of the hierarchical model is compared with another element at the same level in terms of relevance with respect to an element belonging to the upper level. Preferences on the mutual importance of elements are commonly expressed using the 1–9 Saaty's rating scale, according to which a value of 1 indicates that the two considered elements have equal relevance, while a value of 9 means that one element is extremely more important than the other. Even if all AHP models have the main goal in the first level and the alternatives in the last level, it is possible to use several levels for the criteria. A noteworthy example of such multi-level hierarchies introduces a level in which the stakeholders related to the decision problem are explicitly considered, thus enabling multi-actor decision analyses. Several studies are available in the literature that describe the adoption of the AHP method to evaluate a variety of transport-related problems, also with reference to a multi-actor context (cf. [38,39]).

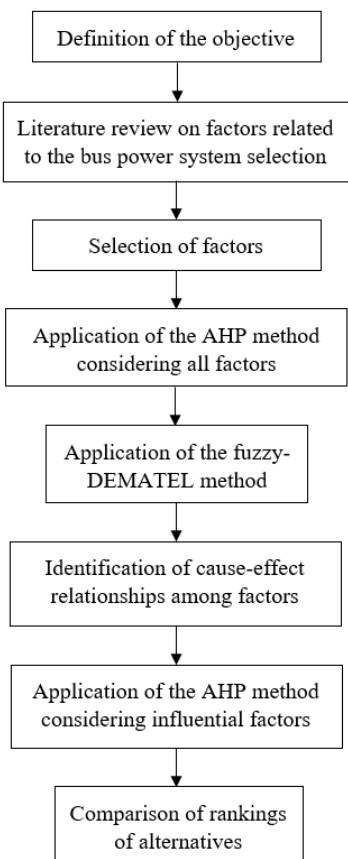

**Figure 3.** Workflow of the development stages of the suggested methodology.

As underlined in [40], the AHP method is suitable for decision problems in which a limited number of evaluation criteria are considered due to the fact that assessing alternatives through many criteria would require a large number of pairwise comparisons, creating redundancy issues. In this regard, the integration of the AHP method with other appraisal techniques, like, for example, the DEMATEL method in the methodology proposed here, contributes to overlook such drawback, relieving the fatigue experienced by respondents and facilitating the resolution of decision-making problems. Indeed, the AHP is one of the evaluation techniques that is most commonly used in combination with the DEMATEL method [21,41,42], whose effectiveness has been proved in many scientific contributions with applications to various economic sectors [16–26]. This method represents a valid approach for obtaining a visual structural model of the casual relationships among factors when solving complex real-world decision-making problems, like choosing the solution for power supply technology for buses. Because human judgments can often be biased and imprecise, the fuzzy logic was used in this study for the implementation of the DEMATEL method to convert linguistic judgments into figures and, thus, to face uncertainty. Notably, in line with the description of the theoretical principles of the conventional and fuzzy version of the DEMATEL method reported in [12], the methodological steps applied on the selected factors are illustrated below:

- Step 1: Selection and involvement of experts. Experts with various expertise related to public passenger transport operations and planning and to electric transport technologies have been selected and were actively engaged in the evaluation process, covering areas of knowledge like operations management, transport planning, vehicle maintenance and repair, and electric vehicle technologies. Notably, the expert panel included actors from different countries such as Italy and Ukraine. These countries present quite distinct economic and political situations, but they both share the global controversial problem of reducing the environmental impacts generated by transport.

No incentives have been provided to the experts to participate in the evaluation procedure, since they have contributed on a voluntary basis considering their interest for the examined decision problem. However, they have been provided with material feedbacks concerning the results of the evaluation procedure, in line with the participatory nature of the adopted approach.

- Step 2: Definition of a linguistic and fuzzy numerical assessment scale. As reported in Table 1, linguistic indicators corresponding to different levels of influence have been associated with numerical values using a triangular distribution, as shown in Figure 4.

**Table 1.** Linguistic indicators and corresponding numerical values.

| Linguistic Indicators | Numerical Values |
| --- | --- |
| Very High (VH) | (0.75, 1.0, 1.0) |
| High (H) | (0.5, 0.75, 1.0) |
| Low (L) | (0.25, 0.5, 0.75) |
| Very Low (VL) | (0, 0.25, 0.5) |
| No Influence (NI) | (0, 0, 0.25) |

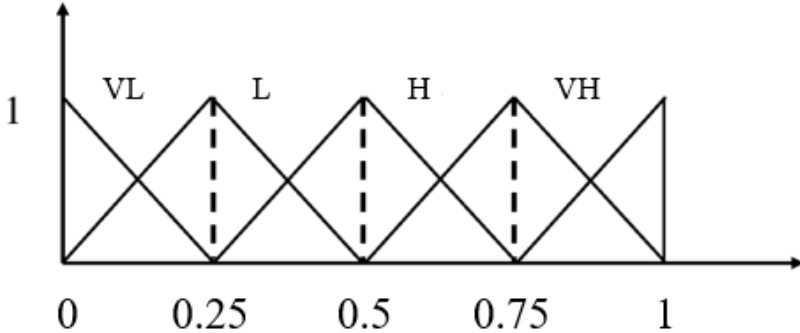

**Figure 4.** Triangular distribution used to convert linguistic indicators into fuzzy numbers [11].

- Step 3: Collection of judgments. Involved actors have been asked to express judgments on the mutual influence of selected factors in linguistic terms, obtaining a judgment matrix for each of them.
- Step 4: Conversion of experts' judgments into fuzzy numbers. Experts' judgments are then converted into fuzzy numbers using the mentioned triangular distribution, obtaining a matrix whose elements indicate the degree of influence of a single factor on the others.
- Step 5: Implementation of the fuzzy DEMATEL method. After creating the corresponding matrix for each considered factor, the average matrix, called the direct dependency matrix, is calculated and then normalized. After further computations a fuzzy general dependency matrix was obtained, which was used to identify and analyze the causal relationships among factors.
- Step 6: Creation of the cause–effect relationship diagram. The sum of the rows and columns of the fuzzy general dependency matrix are denoted as, respectively, vector $\widetilde{D}_i$ and vector $\widetilde{R}_i$ (where $i$ indicates the generic factor). Data contained in these vectors are functional to the creation of the cause–effect relationship diagram, since its horizontal axis, denoted as $(\widetilde{D}_i + \widetilde{R}_i)$, determines the importance of the criterion, while its vertical axis, denoted as $(\widetilde{D}_i - \widetilde{R}_i)$, indicates the degree of mutual influence among factors. Such representation effectively provides useful insights that can contribute to problem solving.

Despite the presence of various scientific contributions using the combination of the DEMATEL and AHP methods, to the best of the authors' knowledge no previous investigation has been carried out by applying these two techniques for the selection of technologies employed in buses for public transport. This research gap has motivated the development of the proposed methodology and its application to a specific case study, which is described in the following section, stressing the context-sensitive aspects that characterized the evaluation procedure.

## 4. Case Study

The combined assessment methodology suggested in this study has been applied to the case study of the city of Trieste, which is a medium-sized Italian city located at the border with Slovenia, with a population of 200.000 inhabitants. The presence of many hills defines the singular morphological configuration of the city, which is responsible for the limited modal share of non-motorized transport solutions. In such circumstances, local public transport plays a key role in urban mobility, not only for commuters but also for the other citizens. The transport offer comprises 56 lines of traditional bus services and one tram line, along with maritime connections during the summer season. In addition, dedicated services (for instance, on-demand and park and ride services) have been experimentally tested in the recent past to meet specific user needs. In general, the public transport service shows great capillarity both in terms of space and time, and it is characterized as being of a high-quality level. The fleet consists of 273 buses and six trams, with a total mileage of 113 million km per year. The service is carried out by a public-private company with almost 800 employees on the basis of a service contract with the public authority, which determines operational goals and constraints, and which establishes the revenue scheme and thresholds [43].

Referring to Figure 3, prior to the actual implementation of the two considered evaluation techniques, the list of factors reported in Table 2 was formed according to the results of the literature review [16–26] and to the authors' experience.

Given the territorial features of Trieste described above, it appears clear that criticalities could be encountered when dealing with the selection of the most adequate solution for the bus power supply technology especially with respect to factors concerning transport, environmental, and operational aspects, such as route length, type of battery and charging station, passenger capacity, and operating speed. Of course, the consequences of such choices on the social and economic aspects are not negligible. Therefore, the proposed evaluation methodology has revealed the relative benefits of the two techniques, with the aim of supporting the local transport company in the decision-making process for the selection of the best bus power supply solution.

The selected factors were used as evaluation criteria to perform an AHP-aided evaluation with the aim of ranking a few alternatives which, as illustrated in Figure 5, consisted of ICEVs, electric vehicles, and hybrid vehicles. Such alternatives represent the most common solutions for urban public transport and range from traditionally fueled buses to fully environmentally sustainable buses. As indicated in the third level of the hierarchical model, the criteria are grouped into macro-criteria (second level) according to the categories defined in Table 2, serving the goal of selecting the best power supply solution for buses. The experts involved in the procedure assessed the relative importance of each element of the hierarchy during some structured interviews by means of pairwise comparisons between elements of the same level. Judgments were stated using Saaty's 1–9 rating scale [44], except for the performances of the alternatives that were evaluated according to a 1–10 scale (the value 1 was associated with the worst performance, while the value 10 was associated with the best performance). Data gathered through the surveys were synthesized and implemented in the model while considering an identical weight for all experts, since they have been assumed to equally contribute to the evaluation process despite their different expertise. The consistency of judgments was ensured by checking that the consistency ratio of all pairwise comparison matrices was smaller than 0.1.

**Table 2.** Factors affecting the selection of the bus power supply technology.

| Category | Factors |
|---|---|
| Social factors | A1. Implementation of modern transport technologies and modern rolling stock. |
| | A2. Driver salary. |
| | A3. Driver satisfaction with working conditions in the transport company. |
| | A4. Control over the psychophysiological state and physical health of the driver. |
| | A5. Culture of work organization at the transport company. |
| Transport factors | A6. Passenger flow. |
| | A7. Route length. |
| | A8. Availability of modern charging stations. |
| | A9. Arrangement of special lanes for bus transit. |
| | A10. Giving priority to bus transit in case of traffic congestion. |
| Environmental factors | A11. Zero emissions. |
| | A12. Availability of technology for the disposal of spent batteries. |
| | A13. Heat, light, noise, and electromagnetic pollution during bus movement. |
| | A14. Climatic conditions for bus operations. |
| | A15. Use of maintenance and repair technologies to ensure an adequate level of environmental safety of the buses. |
| | A16. Type of battery. |
| Operational factors | A17. Bus power reserve on a fully charged battery. |
| | A18. Passenger capacity. |
| | A19. Operating speed. |
| | A20. Ergonomics of the driver workplace. |
| | A21. Ergonomics of the passenger cabin. |
| Economic (financial) factors | A22. Fare. |
| | A23. Payback period of the investment project. |
| | A24. Economic and monetary stimulation to low-income consumers for traveling on environmentally sustainable buses. |
| | A25. Financial losses for the maintenance of the rolling stock. |
| | A26. Loyal financial programs for the transport company for the renewal of the rolling stock. |

Carrying out the AHP evaluation with a wide set of criteria required performing a great number of pairwise comparisons, which implied a high workload for the involved experts. At the same time, such an assessment procedure suggested the possibility that some criteria could depend on a more limited number of them, revealing potential redundancy issues. Consequently, by engaging the same panel of experts, the fuzzy DEMATEL method was employed to identify cause–effect relationships among factors and, thus, highlight the most influential one. As shown in Figure 6, only these latter factors were considered in the second AHP assessment, adopting an approach analogous to the previous procedure. Such a reduction in the number of evaluation criteria has definitely contributed to alleviating the fatigue experienced by respondents while expressing their judgments in the first assessment, since the combinations of possible pairwise comparisons among criteria have been significantly decreased.

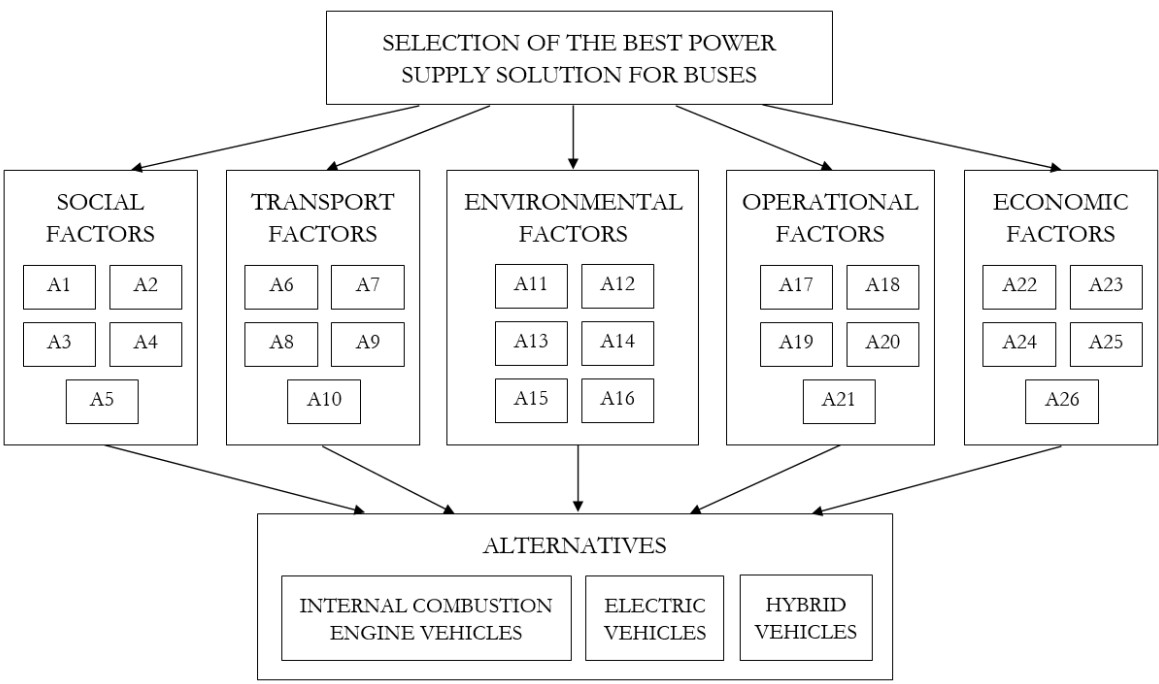

**Figure 5.** AHP hierarchical model including all factors.

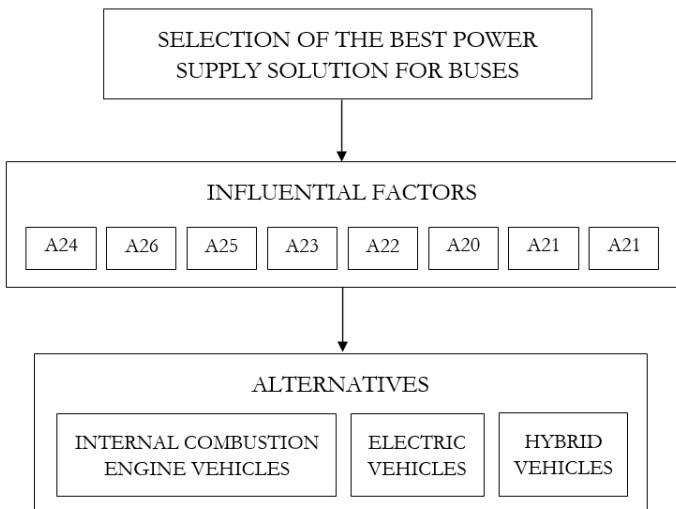

**Figure 6.** AHP hierarchical model including influential factors resulting from the fuzzy DEMA-TEL method.

Details of the results and their discussion are illustrated in the following sections.

## 5. Results and Discussion

Referring to Figure 7, the results obtained from the AHP model including all factors showed that, as far as the macro-criteria are concerned, transport factors have the greatest priority along with operational factors since they encompass practical aspects which directly influence the selection of the bus power supply technology. The evaluation proved that economic factors are also important, given the relevance of ensuring the financial sustainability of interventions. In contrast, less significance is attributed to both environmental and social factors, although they represent two critical aspects that have recently gained increasing attention when planning transport projects and policies.

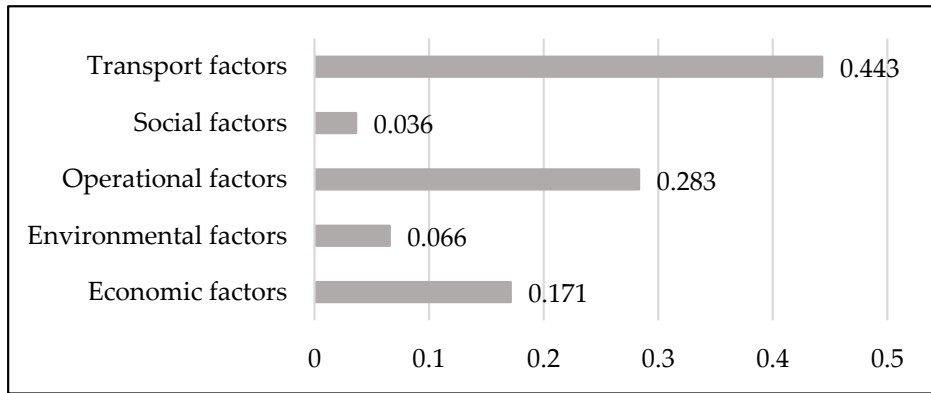

**Figure 7.** Macro-criteria priorities.

The assessment procedure also provided insights into the relative priority of the criteria connected to each macro-criterion. Regarding social factors, Figure 8 indicates that the implementation of modern technologies and rolling stock (A1) has the greatest importance because it is critical to offer a high-quality transport service to all users. Even the culture of work organization at the transport company (A5) proved to be a remarkable factor, as it contributes to the efficiency of the provided service. However, with respect to the goal of selecting the best power supply technology, less significance is attributed to the general working conditions of the driver (A4, A2, and A1), although control over his/her health cannot be overlooked.

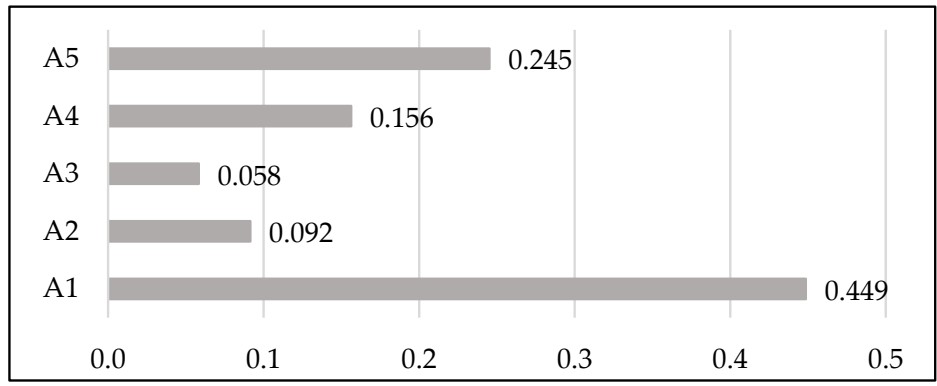

**Figure 8.** Criteria priorities related to social factors.

As reported in Figure 9, among the considered transport factors, the route length (A7) turned out to be a very important criterion, since its determination greatly depends on the vehicle autonomy which in turn is influenced by the type of power supply technology. The passenger flow (A6) is also a relevant criterion in the selection of the bus power supply technology, because the performances of this latter factor need to guarantee the accommodation of the demand, while the availability of charging stations (A8) is essential to ensure a seamless service provision. Criteria concerning the presence of dedicated lanes for buses (A9) and the priority for bus transit in case of traffic congestion (A10) proved to be less significant in the selection of the power supply technology as they are mainly related to mobility planning decisions.

With respect to environmental factors, Figure 10 displays that the criteria concerning the elimination of emissions (A11), and the availability of technology for the disposal of spent batteries (A12) turned out to have the highest importance, since they represent remarkable features in the view of the ultimate goal of environmental sustainability when selecting the power supply technology. Also, the criterion related to climatic conditions (A14) proved to be important, because they directly influence the performances of the traction system, especially in case of batteries in electric vehicles. Less significance is

attributed to maintenance and repair technology requested for environmental safety (A16) as well as to other polluting aspects since they are inherent characteristics of vehicles which may impact the selection of the power supply technology to a lesser extent.

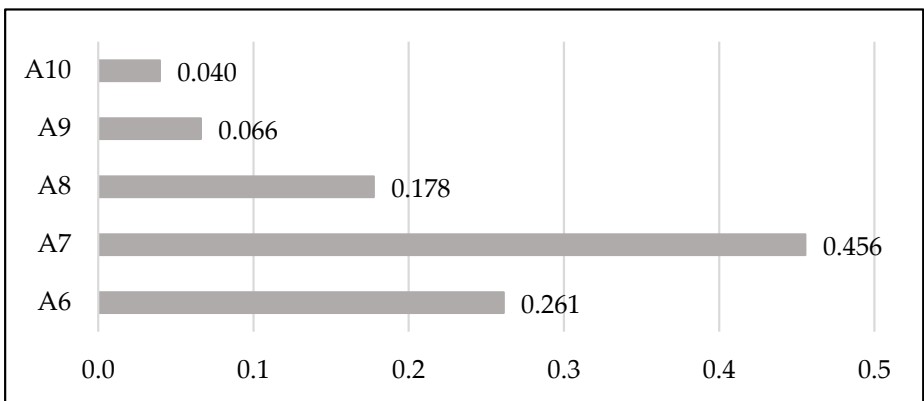

**Figure 9.** Criteria priorities related to transport factors.

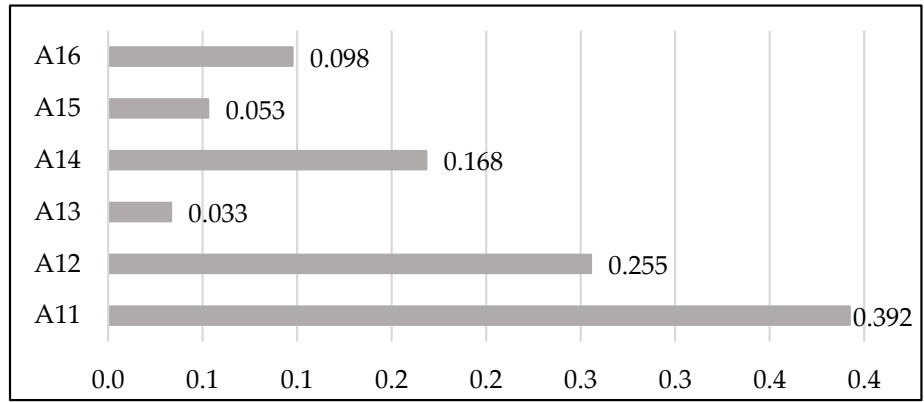

**Figure 10.** Criteria priorities related to environmental factors.

As far as operational factors are concerned, Figure 11 illustrates that criteria related to operational speed (A19), passenger capacity (A18), and bus power reserve to fully charged battery (A17) proved to be the most relevant ones, since they greatly and directly affect the operation of the service and, thus, they are fundamental when choosing the bus power supply technology. On the contrary, the criteria regarding the ergonomics of both the driver workplace (A20) and of the passenger cabin (A21) are much less important.

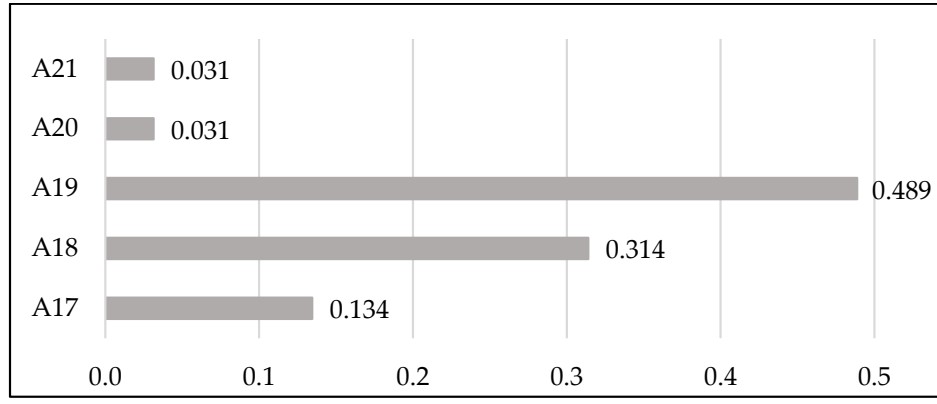

**Figure 11.** Criteria priorities related to operational factors.

With respect to economic/financial factors, Figure 12 reveals that the most important criterion is the one related to the payback period of the investment project (A23), since it is essential for the financial sustainability of the transport company, followed by the presence of loyal financial programs for the fleet renewal (A26), which can actually boost a change in the bus power supply technology. Furthermore, the results coming from the evaluation procedure proved that the ticket fare (A22) plays a relevant role at the financial level, because it represents one of the main sources of revenues of the transport operator and its definition also depends on the selection of the bus power supply technology. In this regard, users' willingness to pay for more environment-friendly power technologies should be analyzed in detail. Less relevance is associated with financial losses for the maintenance of vehicles (A25), which are also strictly connected to the implemented technology and also to the monetary stimulus for low-income consumers (A24), for whom economic convenience tends to be more valuable than environmental sustainability.

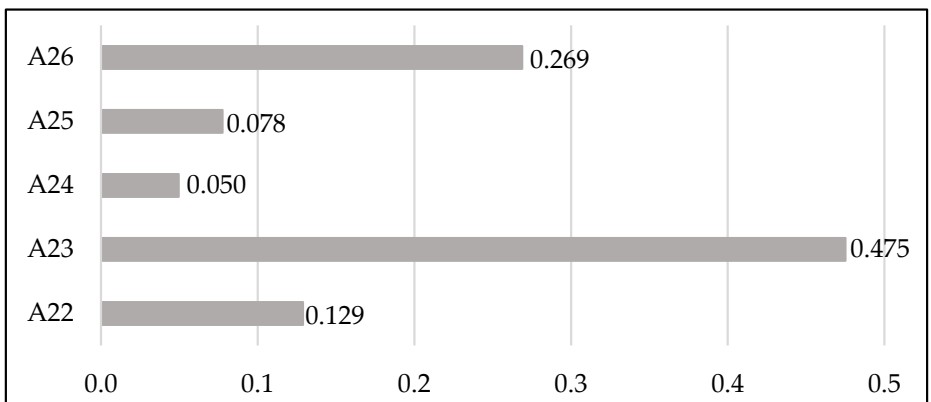

**Figure 12.** Criteria priorities related to economic/financial factors.

The combination of priorities related to macro-criteria and the respective criteria with the performances of the alternatives enabled the aggregation of data concerning experts' judgments, leading to an overall ranking of the examined solutions for bus power supply technology. As shown in Figure 13, internal combustion engine buses proved to be the best alternative with respect to the power supply technology, followed by hybrid buses and electric buses. This is motivated by the fact that internal combustion is the simplest and most diffused technology for buses, and for road vehicles in general, which implies less operational hindrance, and, therefore, a more limited impact from a financial perspective. The reason for the slight difference between the remaining two alternatives can be found in the fact that the technology embedded in hybrid buses is more similar to the one characterizing internal combustion engine buses and thus favors its eligibility. Despite the greater contribution of hybrid and electric buses to a shift towards green mobility, the poor relevance associated with environmental sustainability determined the predominance of ICEVs since no remarkable compensation effects occurred.

As anticipated in the methodological sections, given the high workload to perform pairwise comparisons and the possible redundancy issues in criteria experienced in the AHP evaluation considering all factors, the fuzzy DEMATEL method has been adopted to identify cause–effect relationships among factors and, consequently, to limit their number to the most influential ones. The results of such an application are reported in Table 3, which indicates the significance of each factor (column D + R) and the relationship among them (column D − R). With regard to this latter index, positive values correspond to influential factors, whereas negative values correspond to factors that are influenced by other factors. Based on such data, a rating of the selected factors was defined and a cause–effect relationship diagram was created, as illustrated in Figure 14. The causes are displayed in the chart by blue dots, whereas the effects are represented by red dots.

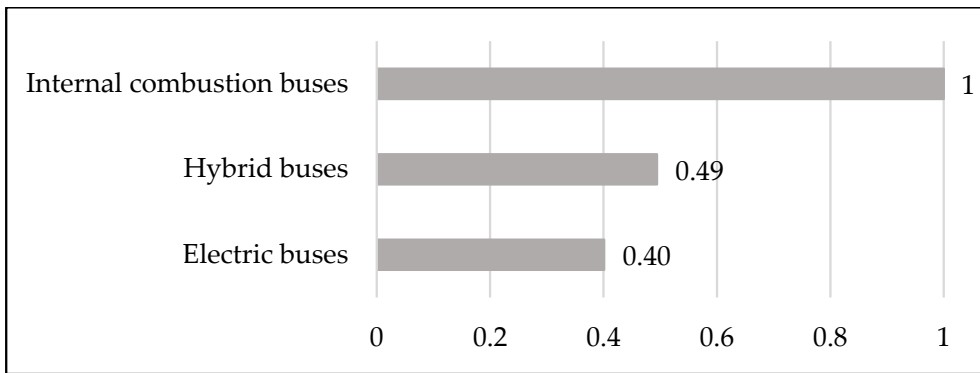

**Figure 13.** Ranking of alternatives considering all factors.

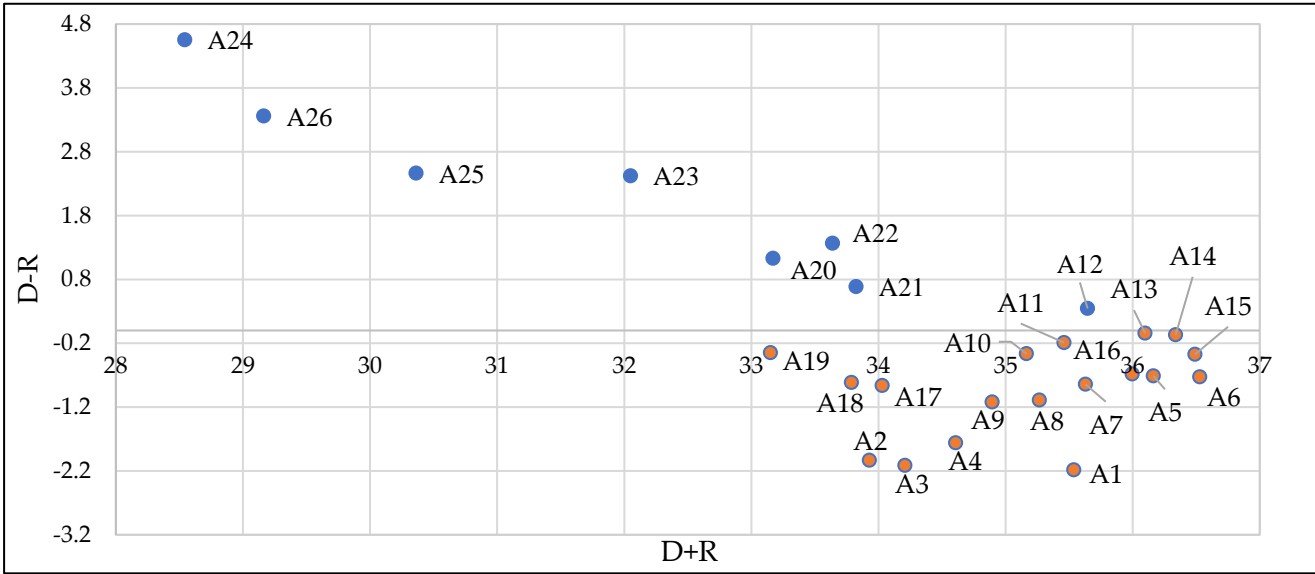

**Figure 14.** Cause–effect relationship diagram of selected factors.

Table 4 reports the denomination of influential factors based on their degree of influence, whereas Table 5 contains the indication of the factors influenced by the former, enabling a distinction between the causal cluster and the effect cluster. It can be noted that the influential factors proved to be related mainly with economic/financial aspects, referring to both capital and operational expenses, followed by social factors connected to the ergonomic characteristics of vehicles. Lastly, the results also showed the influence of environmental issues related to the disposal of spent batteries over the remaining factors. As illustrated in Table 5, the cause–effect relationship analysis performed using the fuzzy DEMATEL method revealed that some of the factors that are most impacted by influential factors regard social and environmental aspects. Notably, they are linked to the implementation of modern transport technologies and rolling stock, driver satisfaction with working conditions in the transport company, absence of polluting emissions, and type of battery. Furthermore, it can be observed that the factors concerning the payback period of the project and the programs for the renewal of the rolling stock influence the greatest number of factors, highlighting their crucial role in the decision-making process regarding the selection of a power supply solution for buses.

**Table 3.** Rating of selected factors based on their mutual influence.

| Factor | D | R | D + R | D − R | Rating |
|--------|---|---|-------|-------|--------|
| A1 | 16.6767 | 18.8608 | 35.5375 | −2.1841 | 26 |
| A2 | 15.9478 | 17.9811 | 33.9288 | −2.0333 | 24 |
| A3 | 16.0478 | 18.1615 | 34.2093 | −2.1137 | 25 |
| A4 | 16.4234 | 18.1844 | 34.6078 | −1.7610 | 23 |
| A5 | 17.7253 | 18.4382 | 36.1635 | −0.7129 | 16 |
| A6 | 17.9013 | 18.6279 | 36.5291 | −0.7266 | 17 |
| A7 | 17.3932 | 18.2359 | 35.6291 | −0.8426 | 19 |
| A8 | 17.0889 | 18.1784 | 35.2673 | −1.0896 | 21 |
| A9 | 16.8877 | 18.0081 | 34.8958 | −1.1204 | 22 |
| A10 | 17.4005 | 17.7641 | 35.1646 | −0.3637 | 13 |
| A11 | 17.6355 | 17.8256 | 35.4612 | −0.1901 | 11 |
| A12 | 17.9956 | 17.6498 | 35.6454 | 0.3458 | 8 |
| A13 | 18.0283 | 18.0699 | 36.0982 | −0.0417 | 9 |
| A14 | 18.1355 | 18.2028 | 36.3383 | −0.0674 | 10 |
| A15 | 18.0580 | 18.4324 | 36.4905 | −0.3744 | 14 |
| A16 | 17.6578 | 18.3396 | 35.9974 | −0.6818 | 15 |
| A17 | 16.5834 | 17.4464 | 34.0299 | −0.8630 | 20 |
| A18 | 16.4876 | 17.3011 | 33.7887 | −0.8135 | 18 |
| A19 | 16.4007 | 16.7501 | 33.1508 | −0.3493 | 12 |
| A20 | 17.1511 | 16.0205 | 33.1716 | 1.1306 | 6 |
| A21 | 17.2558 | 16.5694 | 33.8252 | 0.6864 | 7 |
| A22 | 17.5026 | 16.1371 | 33.6398 | 1.3655 | 5 |
| A23 | 17.2367 | 14.8134 | 32.0501 | 2.4234 | 4 |
| A24 | 16.5479 | 11.9951 | 28.5431 | 4.5528 | 1 |
| A25 | 16.4142 | 13.9487 | 30.3629 | 2.4655 | 3 |
| A26 | 16.2615 | 12.9024 | 29.1639 | 3.3591 | 2 |

**Table 4.** The main influencing factors.

| Rating | Factor |
|--------|--------|
| 1 | A24. Economic and monetary stimulation to low-income consumers for traveling on environmentally sustainable buses. |
| 2 | A26. Loyal financial programs for the transport company for the renewal of the rolling stock. |
| 3 | A25. Financial losses for the maintenance of the rolling stock. |
| 4 | A23. Payback period of the investment project. |
| 5 | A22. Fare. |
| 6 | A20. Ergonomics of the driver workplace. |
| 7 | A21. Ergonomics of the passenger cabin. |
| 8 | A12. Availability of technology for the disposal of spent batteries. |

**Table 5.** Influential and respective influenced factors.

| Influential Factors | Influenced Factors |
|:---:|:---:|
| A24 | A2, A22, A3, A1 |
| A26 | A11, A8, A16, A9, A25, A2, A22, A3, A1 |
| A25 | A1, A15, A16 |
| A23 | A11, A8, A16, A9, A4, A17, A25, A3, A1 |
| A22 | A1, A2, A7, A11 |
| A20 | A19, A4, A3, A1 |
| A21 | A6, A16, A18, A20 |
| A12 | A11, A16, A25, A3, A1 |

The identification of the most influential factors through the application of the fuzzy DEMATEL method served the second AHP evaluation, which benefits from the reduced number of considered criteria. In this case, as displayed in Figure 6, the criteria are not distinguished into macro-criteria because they all belong to the same level of the hierarchical model. The relative priority of each of them is illustrated in Figure 15, which confirms the greatest importance of the criteria related to the payback period of the investment project (A23), the loyal financial projects for the renewal of the rolling stock (A26), and the fare (A22). Even from this assessment procedure, it can be noted that less relevance is attributed to the remaining financial factors and to those concerning environmental (A12) and social (A20 and A21) factors.

As compared to the previous AHP evaluation, Figure 16 shows that similar results were obtained for the ranking of the alternatives, with a more pronounced difference in the preferability of hybrid buses over electric buses given by the less onerous impact of the availability of technology for the disposal of spent batteries and, thus, of the related financial expenses.

The outcomes of the performed evaluation procedure constitute a recommendation for the decision maker, i.e., the transport operator, so additional aspects could potentially be taken into account to define the line of action for the selection of the bus power supply technology, like, for example, the cost of fuel or electricity.

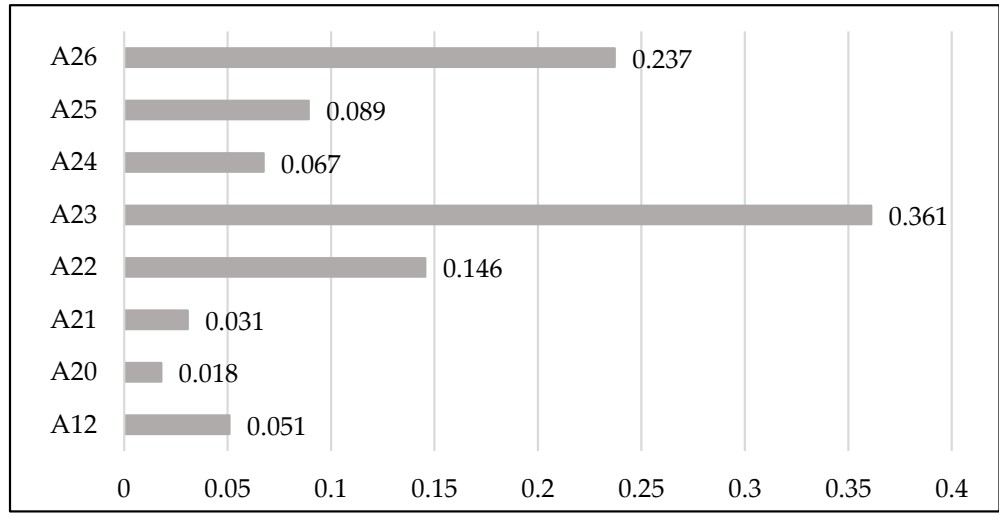

**Figure 15.** Criteria priorities including only influential factors resulting from the fuzzy DEMA-TEL method.

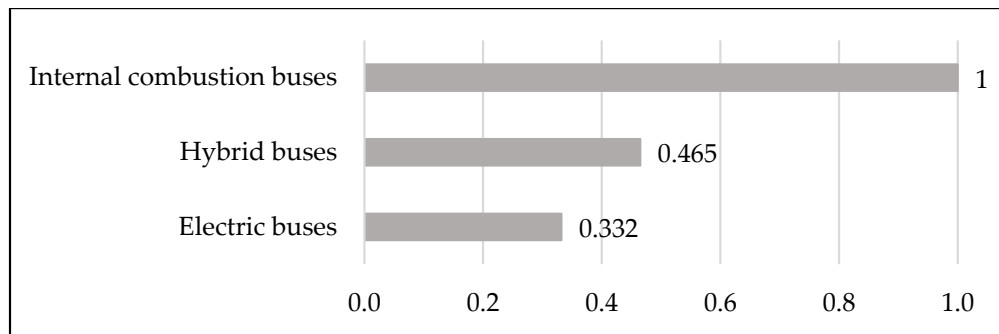

**Figure 16.** Ranking of alternatives considering only influential factors resulting from the fuzzy DEMATEL method.

Finally, a sensitivity analysis was performed to investigate possible variations in the ranking of the alternatives to changing the priorities of criteria. Notably, the sensitivity of the decision model was tested with respect to modifications in the priority of the three most relevant influential factors reported in Figure 15, i.e., those related to the payback period of the investment project, the loyal financial programs for the transport company for the renewal of the rolling stock, and the fare. As illustrated in Figures 17–19, it turned out that no significant modifications in the ranking of the alternatives occur when varying the priority of the criteria, confirming the greater performances of internal combustion buses as compared to the more environmental-friendly solutions. More significant variations in the ranking of the alternatives could be possibly obtained by revisioning the judgments expressed by the experts with respect to criteria priorities, which is part of the future developments of the study.

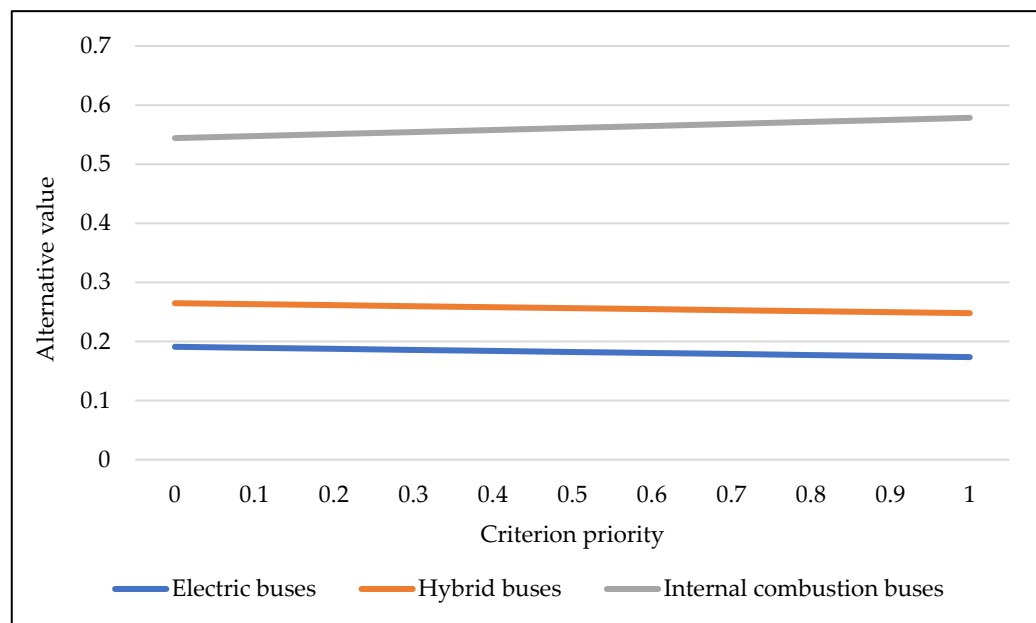

**Figure 17.** Sensitivity analysis for the criterion related to the payback period of the investment project.

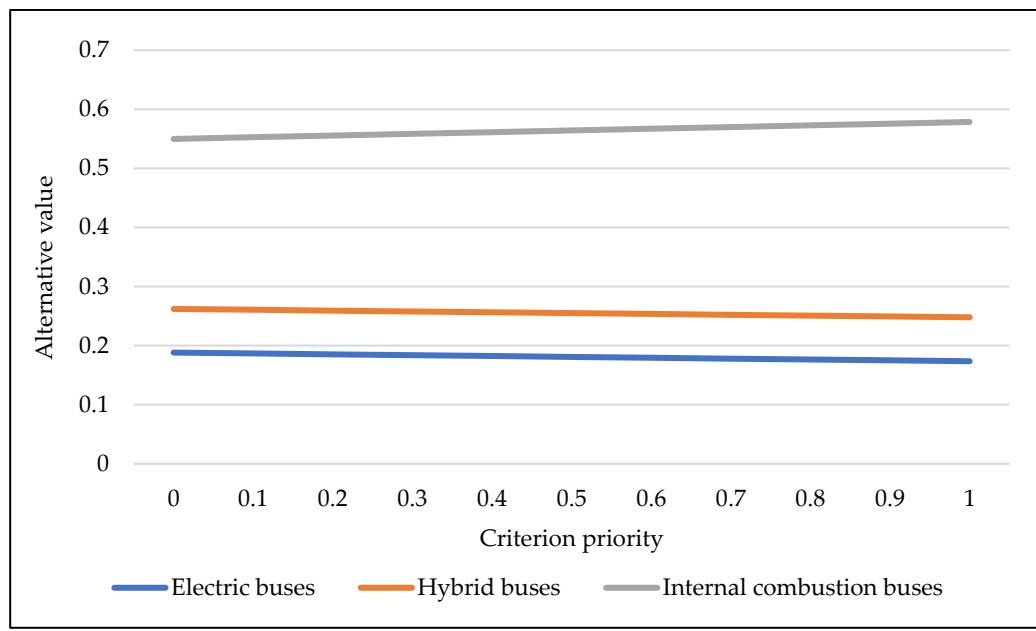

**Figure 18.** Sensitivity analysis for the criterion related to the loyal financial programs for the transport company for the renewal of the rolling stock.

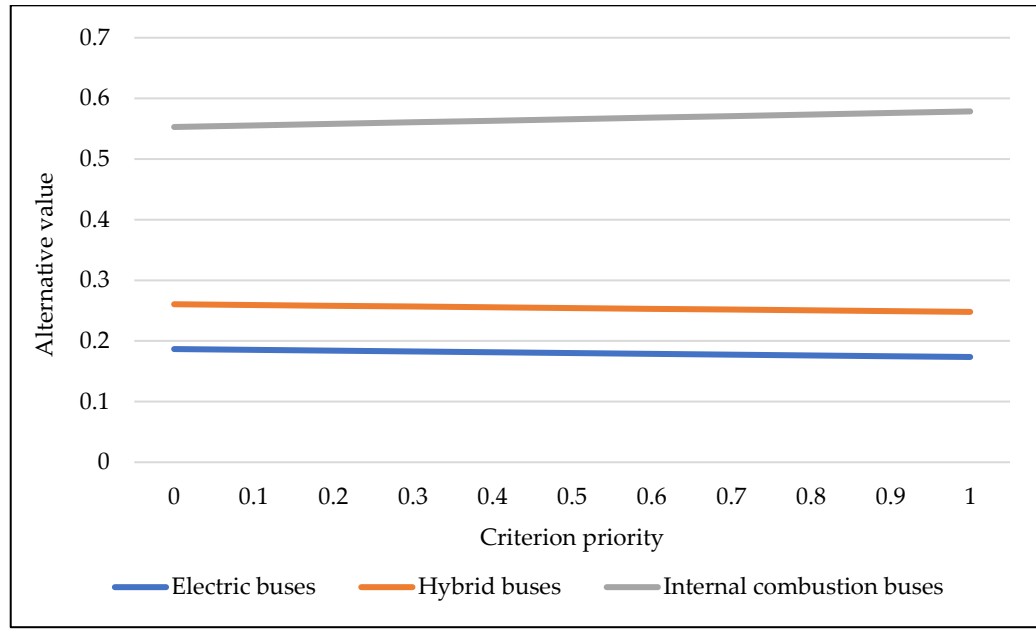

**Figure 19.** Sensitivity analysis for the criterion related to the fare.

## 6. Conclusions

The need to address environmental sustainability in the public transport sector is a key factor in the selection of power supply technology for buses, which represents the main topic of this paper. This decision problem was approached by performing two evaluation procedures, both using the AHP method to rank the alternatives but differing in the number of examined criteria. A wide set of criteria based on a literature review and the experience of a panel of experts was considered in the first AHP model, whereas a reduced set of the most influential criteria was employed in the second AHP model. The reduced set was identified through the application of the fuzzy DEMATEL method, which revealed the cause–effect relationships between the criteria. Other than that, the reduction in the number of the evaluation criteria permitted the alleviation of the fatigue experienced by

experts while responding to pairwise comparisons during the first assessment. Referring to the case study of the city of Trieste, similar rankings were obtained from the two evaluation procedures, suggesting that internal combustion engines represent the best power supply technology for buses compared to electric and hybrid engines. This outcome is motivated by the relevance associated with transport, operational, and economic factors, for which internal combustion engines present fewer constraints. Because of the considerable weights given to these factors, technologies that have better performance from environmental and social viewpoints cannot obtain an overall performance that is sufficiently high to place them in the highest ranks.

The significance of the study presented in this paper consists in the comparison between two rankings of alternatives that have been obtained by considering different dimensions of analysis, i.e., not only the relative importance of evaluation criteria but also their mutual influence. As regards the research field in which investigations have been performed, this research stresses the relevance of making informed decisions in the public transport sector to increase its sustainability.

The implications of the study can be summarized as follows, according to two diverse perspectives:

- At the academic level, the attempt to simplify the evaluation procedure by integrating the AHP and DEMATEL methods to reduce the number of evaluation criteria was challenged by the difference between the concept of importance and influence, which may improperly affect the selection of criteria. As a matter of fact, the most influential criteria may not necessarily correspond to the most important criteria;
- At a practical level, the study emphasized that great attention should be paid to the purpose for the application of the evaluation methods, especially when multiple experts are engaged. Notably, the assessment procedures carried out in the study revealed the need to accurately explain the specific goal of the adopted evaluation techniques to the respondents. In this respect, a clear distinction between the concepts of importance and influence was necessary to implement the AHP and DEMATEL methods correctly. Indeed, as underlined in [45], possible misunderstanding issues characterizing the analyst's point of view can introduce biases in decision-aiding processes, which are then reflected in the final recommendation provided to the decision maker. Referring to the outcomes of the study, the potential misinterpretation of the conceptual difference between the notion of importance and influence may be the reason for the higher rank achieved by internal combustion buses despite the generally diffused awareness towards environmental sustainability.

Future developments of the research will include, on one hand, a deeper investigation on the soundness of the experts' understanding of the evaluation approach and, on the other hand, a more comprehensive sensitivity analysis of the priorities of the criteria in order to detect possible variations in the ranking of alternatives. Furthermore, the transferability of the proposed methodology will be tested on other case studies and contexts.

**Author Contributions:** Conceptualization, G.L., E.P. and C.C.; methodology, C.C. and M.V.; software, C.C. and M.V.; validation, G.L. and E.P.; formal analysis, C.C. and M.V.; writing—original draft preparation, M.V.; writing—review and editing, C.C.; supervision, G.L. and E.P. All authors have read and agreed to the published version of the manuscript.

**Funding:** This research received no external funding.

**Institutional Review Board Statement:** Not applicable.

**Informed Consent Statement:** Not applicable.

**Data Availability Statement:** Data are contained within the article.

**Conflicts of Interest:** The authors declare no conflict of interest.

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
