# Peer review of "Comparing Power Supply Technologies for Public Transport Buses through the AHP and the Fuzzy DEMATEL Method"

_sustainability, doi:10.3390/su152316190_

Round 1

Reviewer 1 Report

Comments and Suggestions for Authors

1. Write proper introductions with recently published papers.

2. Elaborate your method 

3. Check all the results and add more results to validate your claim

4. Modify abstract and conclusion both.

5. Methodollogy and case study not clear 

Comments on the Quality of English Language

Please improve

Reviewer 2 Report

Comments and Suggestions for Authors

In this paper, the authors have Power supply Technologies for Public Transport 2 Buses through the AHP and the Fuzzy-DEMATEL Method. These are my concerns on the paper.

1. The novelty of the paper is not clear.

2. Why authors choose AHP and Fuzzy DEMATEL method? The reason was not clear.

3. How are the criteria shown in Fig. 7 to Fig. 13 obtained? 

Comments on the Quality of English Language

The language of the paper is good. 

Reviewer 3 Report

Comments and Suggestions for Authors

Dear Authors,

it was a pleasure reading your paper. The net zero emission goal is a very ambitious one. The GHG emissions leading to global warming must be reduced, and as around 25% of the emissions comes from transport it's important to find greener solutions in this sector. Your paper focuses on an important decision transport companies need to make in order to be more sustainable.

However, some issues remain that need your attention:

# in lines 158, 187 there is a different font size compared to the rest of the paper

# Funding, Institutional Review Board Statement, Informed Consent Statement, Data Availability Statement, Acknowledgments and Conflicts of Interest are not stated

With regards,

Reviewer

Reviewer 4 Report

Comments and Suggestions for Authors

In general, this article is very creative, both in terms of research topic and research methods, but there is still some distance from the quality that can be published.

1. I think that in the introduction, it is essential to provide reasons for using AHP and DEMATEL as the two research methods in this study are not to be overlooked. Furthermore, this chapter should emphasize the significance, necessity, research gap, and primary objectives of this study.

2..In the literature review, it is necessary to supplement a substantial amount of literature. Currently, the quantity of literature and the content covered are insufficient for journal publication. I recommend supplementing the literature from the following five sections: public transport, bus power supply technology, AHP, fuzzy-DEMATEL, and its development background.

3. Please clearly label the sources and origins of all 16 figures. If they are based on or inspired by other research, please provide authorization or proof of permission for each figure.

4. Please explain the characteristics of the respondents and how to avoid respondents experiencing fatigue while answering.

5. One very regrettable aspect in the end is the lack of research significance, research recommendations, research contributions, and research limitations~

Comments on the Quality of English Language

There is room for improvement in English writing for academic articles

Author Response

-

Round 2

Reviewer 4 Report

Comments and Suggestions for Authors

1. What is the reference source for Figure 2? Is there authorization for its use(Citing a source and being authorized are not the same)?

 2.In a more detailed manner, emphasizing the academic and practical implications is a crucial component of a journal article. Therefore, I recommend reevaluating and supplementing this section.

3.Do you provide participants with material feedback or incentives?

Comments on the Quality of English Language

Minor editing of English language required:

Please pay attention to language coherence and accuracy
